# Differences in the Moisture Capacity and Thermal Stability of *Tremella fuciformis* Polysaccharides Obtained by Various Drying Processes

**DOI:** 10.3390/molecules24152856

**Published:** 2019-08-06

**Authors:** Chun-Ping Lin, Shu-Yao Tsai

**Affiliations:** 1Department of Food Nutrition and Health Biotechnology, Asia University, 500, Lioufeng Rd., Wufeng, Taichung 41354, Taiwan; 2Office of Environmental Safety and Health, Asia University, 500, Lioufeng Rd., Wufeng, Taichung 41354, Taiwan; 3Department of Medical Research, China Medical University Hospital, China Medical University, 91, Hsueh-Shih Rd., Taichung 40402, Taiwan

**Keywords:** polysaccharide, *Tremella fuciformis*, moisture capacity kinetic, thermal effect, drying process

## Abstract

We compared the proportions and differences in the polysaccharides of *Tremella fuciformis* (Berkeley) after drying them by various processes, such as 18 °C cold air, 50 °C hot air, and freeze-drying. We also focused on the moisture capacity kinetic parameters of *Tremella fuciformis* polysaccharides using various thermal analyses, including differential scanning calorimetry and thermogravimetric techniques. Erofeev’s kinetic and proto-kinetic equations, utilized for kinetic model simulation, can predict the moisture capacity due to the thermal effect. Among the various drying processes, cold air-drying had the highest molecular weight of 2.41 × 10^7^ Da and a moisture content of 13.05% for *Tremella fuciformis* polysaccharides. Overall, the freeze-dried products had the best thermal decomposition properties under the conditions of a closed system, with an air or nitrogen atmosphere, and had an excellent moisture capacity of around 35 kJ/kg under a closed system for all samples.

## 1. Introduction

*Tremella fuciformis* (*T. fuciformis*) (Berkeley) contains high amounts of carbohydrates, proteins, and vitamins, and a variety of amino acids [1]. In particular, plentiful polysaccharides are the main bioactive ingredients of *T. fuciformis* [2,3,4]. *T. fuciformis* polysaccharides are mainly composed of a linear (1→3)-linked *α*-mannose backbone, with mostly β-xylose and β-glucuronic acid [5,6,7]. The *T. fuciformis* polysaccharide molecular weight is approximately 5 × 10^5^ to 6 × 10^6^ Da, making it an acidic heteropolysaccharide. The structure is very different from that of black fungus and other mushrooms that contain *β*-(1,3)-D-glucan as the same homopolysaccharide of their main structures [8,9,10].

According to the literature, *T. fuciformis* polysaccharides have antioxidant [8,11], anti-inflammatory [12], immunomodulating [13], anti-fatigue [14], anti-radiation [15], and anti-diabetic activity [16]. In addition, the *T. fuciformis* polysaccharide, when coated on the skin, can form a transparent film that is able to improve its water retention rate. This results in an anti-wrinkle and natural moisturizing material, which is suitable for the development of skin moisturizing or skin-care products that can also be used to protect the skin from the ultraviolet light that causes aging due to sunlight [17,18]. The fresh fruit body of *T. fuciformis* is difficult to store because browning, odor change, and deterioration can occur, even in a refrigerated environment.

How to preserve the active ingredients and control the flavor is an issue of concern in the industry. In particular, *T. fuciformis* has a very high polysaccharide content among edible and medicinal mushrooms. The aim of this study involved comparing the proportions and differences in the composition of *T. fuciformis* polysaccharides produced by various drying processes, such as freeze-drying, 18 °C cold air-drying, and 50 °C hot air-drying. We also focused on the endothermic reaction kinetic parameters of *T. fuciformis* polysaccharides using various thermal analyses, including differential scanning calorimetry (DSC) [19] and thermogravimetry (TGA) [20] techniques, with which we could ascertain their characteristics as additives so that they can be added to cosmetics, health foods, biomaterials, and other products.

We conducted a micro-thermal analysis to discover the moisture content in biomaterials, which can be applied in the processing of medical or health foods [19]. This biotechnology is very important as it affects the active and storage conditions of biomaterials [21]. In addition, a high moisture content also increases the risk of microbial contamination, and thus, it is important to control the moisture and then avoid the interference or metamorphic factors of the biomaterial processing to ensure product quality and safety. Herein, the analyses involved using a closed system, filled with an air or a nitrogen atmosphere, which also allowed the differences in moisture desorption to be obtained, and these could be applied to the environmental control of the conditions in the processes.

The endothermic and mass loss kinetic evaluation of moisture desorption belongs to a special physical phenomenon or change that depends on the temperature shift [19,20,21,22], which is never associated with a chemical reaction. The moisture desorption of *T. fuciformis* polysaccharides is related to the molecular weights and the drying process. We compared two kinetic models: Erofeev’s kinetic equation and the proto-kinetic equation [19,22]. Two types of thermal analysis models obtained by DSC and TGA tests were produced for moisture desorption kinetic simulation and differences in the environmental atmosphere, which can predict the moisture desorption under heating. Following this, we determined the processing and application conditions for *T. fuciformis* polysaccharides.

## 2. Results and Discussion

### 2.1. Basic Characteristics of Various Drying Processes for T. fuciformis Polysaccharides

From Table 1, when evaluating the color parameter of the *T. fuciformis* product dried by various processes, for the freeze-dried sample, a product brightness *L** value of 79.33 was obtained, which is higher than the others, and the *a** value of 7.79 and *b** value of 16.02 are lower than the values for the other drying processes. Overall, among the products, the freeze-dried sample WI value of 72.71 was higher than that of the samples from the other drying processes, and the hot air-dried sample was the darkest among all samples. Figure 1 shows the products of various drying processes; the freeze-dried sample is the brightest, and the hot air-dried sample is the darkest brown. In addition, Figure 1 displays the products produced when drying *T. fuciformis* polysaccharides by various processes; for the hot air-dried polysaccharide, the color is also the darkest brown among all of the samples.

Moreover, Table 2 presents the preparation of polysaccharides in this study; from the results of basic characteristic analyses of *T. fuciformis* polysaccharides dried by various processes, we obtained a yield rate, molecular weight, and moisture content above 15.35%, 2.06 × 10^7^ Da, and 11.61% for freeze-dried *T. fuciformis* polysaccharides, respectively. Among the various drying processes, the cold air-drying had a high molecular weight 2.41 × 10^7^ Da and moisture content of 13.05% for *T. fuciformis* polysaccharides. In addition, for the products of 18 °C cold air-, 50 °C hot air-, and freeze-drying processes, the freeze-dried product was fluffy and light, but the 18 °C cold air- and 50 °C hot air-dried products were hard and brittle, with a dark brown color (Figure 1). After grinding, the 18 °C cold air- and 50 °C hot air-dried products displayed a grainy texture as a food additive, but freeze-drying had a less significant effect.

### 2.2. Tremella fuciformis Polysaccharides Moisture Desorption and Thermal Decomposition Properties

*Tremella fuciformis* polysaccharides dried by various processes were characterized for obtaining the temperature and enthalpy of moisture capacity and thermal stability under a closed system using DSC. From Figure 2, the DSC tests show that the polysaccharides dried by various processes were evaluated using a scanning rate of 6 °C/min (see Figure 2a), and this was selected by evaluating the freeze-dried sample using various scanning rates of 6, 8, and 10 °C/min (see Figure 2b) for the moisture capacity and thermal stability under closed system DSC analyses. Table 3 shows the moisture capacity and thermal stability using a scanning rate of 6 °C/min, and for the freeze-dried, cold air-dried, and hot air-dried *T. fuciformis* polysaccharides, the onset temperature of moisture desorption (en*T*_o_) and their peak maximum temperature (en*T*_p_) was 56.27, 41.67, and 41.68 °C and 107.84, 89.42, and 88.32 °C for the moisture desorption, respectively.

In addition, the freeze-dried, cold air-dried, and hot air-dried polysaccharides at the selected scanning rate of 6 °C/min had enthalpy of moisture capacity (enΔ*H*) values of 362.70, 321.90, and 253.50 kJ/kg, respectively. Furthermore, from Figure 2a and Table 3, among the freeze-dried, cold air-dried, and hot air-dried polysaccharides, the polysaccharides of the freeze-dried sample had higher en*T*_p_ and enΔ*H* values compared to the others. It is obvious that the freeze-dried polysaccharides have a better moisturizing capability, which is indicated by the ability of the closed system to absorb moisture by DSC analysis. In particular, comparing Table 1 and Table 3 indicates that the cold air-dried polysaccharides have a high molecular weight and moisture content, but are not the most difficult sample to use in moisture desorption, according to the DSC tests in this study.

Regarding the thermal decomposition stability, there are only small differences in the ex*T*_o_ and e*xT*_p_ of the *T. fuciformis* polysaccharides dried by various processes, but the freeze-dried polysaccharides showed an illustrious increase in exΔ*H*_d_, with values of 108.70, 72.73, and 60.61 kJ/kg for the thermal decomposition reaction by DSC at various scanning rates of 6, 8, and 10 °C/min, respectively. The thermal decomposition behavior because of heat could build an e*xT*_p_ of around 290 °C for all the samples of *T. fuciformis* polysaccharides dried, and this value was very consistent for any of the drying processes. In addition, Table 3 shows the results for *T. fuciformis* polysaccharides dried by various drying processes; when added to cosmetics, health foods, and biomaterials, heat treatments should never be more than approximately 260 °C. In addition, with regards to the moisture capacity and thermal decomposition of *T. fuciformis* polysaccharides dried by various processes under a closed system determined by DSC analysis, the faster the heating rate, the smaller the value.

### 2.3. Thermal Mass Loss of T. fuciformis Polysaccharides

From Figure 3 and Table 4, we obtained the thermal mass loss (mass%) and the differential thermal mass loss of *T. fuciformis* polysaccharides dried by various processes, and estimated the temperatures of moisture capacity and thermal stability analyses by open system TGA tests in nitrogen and air atmosphere. Table 4 shows the results of TGA analyses of the thermal mass loss and the differential thermogravimetric (DTG) curve at heating rates of 6, 8, and 10 °C/min. From Figure 3a,b, we selected the heating rate 6 °C/min for the comparative thermal mass loss analysis of polysaccharides dried by various methods. Table 4 and Figure 3b 4 show the peak maximum temperature (*T*_p_) using differential thermal mass loss by time and the mass loss associated with the moisture desorption and thermal decomposition characteristics of *T. fuciformis* polysaccharides dried by various processes in nitrogen and air atmosphere.

Meanwhile, for the moisture desorption and thermal decomposition stability, the peak maximum temperatures of moisture desorption (md*T*_p_) and thermal decomposition (td*T*_p_) and the mass losses in nitrogen and air atmosphere of the polysaccharides from freezing-, cold air-, and hot air-drying were evaluated by TGA tests. Among the polysaccharides dried by various methods, for the freeze-dried sample, there was a thermal delay effect by multiple heating rates of 6, 8, and 10 °C/min, and the values of md*T*_p_ and td*T*_p_ exhibited linear increases, but the cold air- and hot air-dried samples were non-linear and chaotic.

In addition, as seen in Table 4, the freeze-dried polysaccharides have nice thermal stability characteristics, and the residual non-decomposable substances of mass lost 81.58 and 92.70 mass% (at a heating rate of 6 °C/min), but the cold air-dried sample kept more of the less thermally decomposable substances of mass, losing 74.72 and 75.54 mass% (selected heating rate of 6 °C/min) under nitrogen and air conditions when using TGA tests, respectively. According to Table 4, in terms of the thermal decomposition stability in nitrogen and air atmosphere, for the md*T*_p_, td*T*_p_, moisture of mass loss, and thermal decomposition of mass loss of polysaccharides dried by various methods, the freeze-dried polysaccharides had more stable characteristics, but cold air- and hot air-dried polysaccharides also showed less thermal stability, which was close to the results of the DSC tests.

In Figure 4, *T. fuciformis* polysaccharides from various drying processes under nitrogen have a better stability at the peak maximum temperature of 288 °C (freeze-dried at a heating rate of 6 °C/min), which is greater than under an air atmosphere (274 °C). This shows that *T. fuciformis* polysaccharides in a nitrogen atmosphere exhibit better preservation for storage conditions. In addition, Figure 5 shows interesting preliminary differences in the DSC and TGA analyses; the DTG curve clearly exhibits the mass loss of moisture desorption of ca. 10 mass%. The moisture content of *T. fuciformis* polysaccharides dried by freezing, 18 °C cold air, and 50 °C hot air was 11.61%, 13.05%, and 12.65% (see Table 2), respectively, but DSC tests do not present straightforward information. Here, it could not be confirmed that the moisture content of the polysaccharides is associated with the adsorption capacity.

### 2.4. Moisture Content Stability and Characteristics Obtained by Desorption Kinetic Simulation

The moisture desorption of the *T. fuciformis* polysaccharides dried by various processes belongs to an unsure class of reactions. Erofeev’s kinetic and the proto-kinetic equation were applied to an evaluation of the moisture content stability and characteristics, which could determine the moisture desorption kinetic parameters of endothermic desorption and the moisture loss reaction under a closed system in nitrogen and air conditions, respectively [20,21]. Comparisons of the experimental data and data derived from simulated Erofeev’s kinetic and proto-kinetic equations for the heat production rate or mass loss rate versus time for the conducted DSC tests and TGA analyses of *T. fuciformis* polysaccharides from various drying processes are shown in Figure 6. In contrast to Figure 6, the use of simulated proto-kinetic models gave superior results. Table 5 and Table 6 display the results of the desorption kinetic simulations of *T. fuciformis* polysaccharides dried by various processes.

Table 5 shows that the values of freeze- and hot air-dried kinetic parameters derived from the simulated proto-kinetic model match the results obtained from the multiple scanning rates of 6 and 8, and 10 °C/min, but the cold air-dried sample exhibited inconsistent results in the moisture capacity kinetic parameters by Erofeev’s kinetic and proto-kinetic equation simulations, which also demonstrated that the cold air-dried sample had less moisture desorption stability under the thermal effect. In addition, by comparing Table 3 and Table 5, we determined that the freeze-dried sample had high en*T*_o_, en*T*_p_, and enΔ*H* values among the drying processes, and the *E*_a_ values of moisture capacity were greater than the others, as determined by conducting DSC closed system tests. Even though, as shown in Table 2, the freeze-dried sample has the lowest extraction rate, molecular weight, and moisture content of polysaccharides, this sample has an amazing moisture capacity when considering all of the samples, as determined by DSC tests in this study.

Simultaneously, the analysis of the kinetic parameters of moisture capacity of the *T. fuciformis* polysaccharides from various drying processes depended on the reliability of the kinetic model. In addition, we applied Erofeev’s kinetic and proto-kinetic model simulation for a straight evaluation of the kinetic parameters of *T. fuciformis* polysaccharides from various drying processes under nitrogen and air conditions and compared the results to the simulated moisture capacity produced by TGA tests. From a comparison of Figure 6 and Table 6, the use of simulated proto-kinetic models gave superior results. In particular, the 10 °C/min heating rate is incongruous with the other heating rates. This data set was excluded from further analysis. From Table 6 came the interesting result that the hot air-dried *E*_a_ values were greater than the others under nitrogen and air conditions. Moreover, the cold air-dried sample had lower *E*_a_ values and confusing information compared to the other *T. fuciformis* polysaccharides resulting from various drying processes, which was the same result as for the DSC analysis.

## 3. Materials and Methods

### 3.1. T. fuciformis Samples and Drying Methods

Fresh fruiting bodies of *T. fuciformis* were donated by Wansheng Science and Technology Agriculture Co., Ltd. (Changhua, Taiwan). Fresh *T. fuciformis* fruiting bodies were rinsed, followed by removal of the substrate and slicing to a thickness of 3–4 cm. The experiments were carried out using 2 kg of mushrooms in the cold air-, hot air-, and freeze-drying methods. The cold air drying used a food dehydrator (Excalibur 3926TB, Sacramento, CA, USA) at 18 °C for 96 h. The hot air drying used an oven dryer (DV453; Channel, New North City, Taiwan) at 50 °C for 72 h. The freeze drying used a lab-scale freeze dryer (FD8080; Hansor, Taichung, Taiwan) for freeze drying for 72 h. The drying process lasted until the moisture content of *T. fuciformis* reached below 11% (wet basis) (see Figure 1). After drying, the samples were ground into a fine powder (60 mesh) by a mill (Retsch Ultracentrifugal Mill and Sieving Machine, Haan, Germany).

### 3.2. Dried Products of T. fuciformis Color Measurement

The reflective surface color of the *T. fuciformis* dried product was measured using an S80 color measuring system, and *L*, *a*, and *b* values were recorded by a BYK Gardner SP-6692 instrument (BYK Additives & Instruments, Columbia, MD, USA) at room temperature and 60% relative humidity. A standard white plate (X = 91.98, Y = 93.97, and Z = 110.41) was used to standardize the instrument. The whiteness index (WI) was calculated based on the following equation [23]:(1)$$WI=100-\sqrt{{\left(100-L\right)}^{2}+a^{2}+b^{2}}$$

### 3.3. Preparation of T. fuciformis Polysaccharides

Four grams of *T. fuciformis* powder, by various drying processes, were mixed with 120 mL of water and extracted in an autoclave at 121 °C for 15 min for each experiment. After the extraction, the extract was centrifuged at 3000 rpm for 20 min at room temperature, and the residue and supernatant were separated. The residue for each sample was mixed with 1% sodium carbonate (ratio 1:30 *w*/*v*) at 100 °C for 60 min. After the second extraction, the extract was centrifuged at 3000 rpm for 20 min at room temperature, and the residue and supernatant were separated [19]. The combined twice-extracted supernatant was dialyzed by using a Cellu Sep T2 tubular membrane (MWCO: 6,000-8,000, Membrane Filtration Products, Inc., Seguin, TX, USA) for 24 h. The dialysate was concentrated to a small volume and precipitated with three volumes of 95% ethanol at 4 °C for 24 h. The crude *T. fuciformis* polysaccharide precipitate was separated by centrifugation at 4000 rpm for 20 min [19]. The *T. fuciformis* polysaccharide sample was freeze-dried and ground into the powder (60 mesh) (see Figure 1). The yield was calculated using the following calculation: yield/mass% = Mt/Mi ×100, where Mt is the crude polysaccharide extract and Mi is the mass of the *T. fuciformis* powder.

### 3.4. T. fuciformis Polysaccharide Molecular Distribution Analysis

The molecular weight distribution was measured by high performance liquid chromatography (HPLC) according to the previous method [19]. It was performed on a Hitachi L-2130 chromatograph equipped (Hitachi, Tokyo, Japan) with Polysep-GFC-P 4000 and Polysep-GFC-P 1000 (7.8 × 300 mm, Phenomenex) and a refractive index detector (Hitachi L-2490). The mobile phase was ultrapure water and the flow rate was 1.0 mL/min. The columns were maintained at 30 °C. Samples were prepared at the concentration of 5.0 mg/mL. Different molecular weights of dextran standards (Mw: 180, 1.5 × 10^3^, 6.0 × 10^3^, 4.0 × 10^4^, 1.0 × 10^5^, 9.0 × 10^5^, and 2.0 × 10^6^ Da) were used to establish a calibration curve [19].

### 3.5. T. fuciformis Polysaccharide Moisture Content Measurement

The moisture content of the *T. fuciformis* polysaccharide samples from various drying processes was measured by an MB45 moisture analyzer (Ohaus Corporation, Parsippany, NJ, USA) at 105 °C. Samples of 1 g were used until a constant weight was reached. The results are expressed as the percent loss of polysaccharide moisture content.

### 3.6. Tremella fuciformis Polysaccharide DSC Tests

The moisture capacity and thermal decomposition reaction calorimetric measurements of *T. fuciformis* polysaccharides were conducted with a TA Q20 DSC (TA Instruments, New Castle, DE, USA). The samples were sealed in 20 μL aluminum pans; the test pan was sealed manually by a special tableting device that accompanied TA’s DSC. The carrier gas was 98% nitrogen by processing a flow rate of 50 mL/min for all samples. ASTM E698 was used to acquire the micro calorimetric measurements for analyzing the parameters. Approximately 1.0 to 1.6 mg of the sample was used to acquire the experimental data. The non-isothermal heating rates were selected at 6, 8, and 10 °C/min for the temperature rise range of 30 to 450 °C for each DSC test. [19,21]

### 3.7. T. fuciformis Polysaccharide TGA Analyses

Mass loss dynamic heating experiments to determine the moisture capacity and thermal mass loss of *fuciformis* polysaccharides were carried out on a Mettler-Toledo 2-HT thermogravimetric analyzer (Mettler-Toledo, Schwerzenbach, Switzerland). About 6.0 mg per sample was put in a 70 μL ceramic test pan for each TGA experiment, and the analysis results were applied to collect the experimental data by STAR^e^ software (version 9.0, Mettler-Toledo, Schwerzenbach, Switzerland) [20]. The selected non-isothermal heating rates were from 30 to 500 °C at various rates of 6, 8, and 10 °C/min. The 97% purity nitrogen and air gasses at a flow rate of 60 mL/min were used as the gas atmosphere under the non-oxidation and oxidation conditions of the TGA tests, respectively.

### 3.8. T. fuciformis Polysaccharide Moisture Capacity Kinetic Simulation

Erofeev’s kinetic and the proto-kinetic equation were applied to the model simulation to evaluate the moisture capacity for the desorption reaction or the mass loss of *T. fuciformis* polysaccharides as follows [19,22]:(2)$$r_{i}=ke^{\frac{-E}{RT}}\left(1-\alpha \right){\left[-\ln(1-\alpha )\right]}^{n_{0}}\;\mathrm{Erofeev}’s\;\mathrm{kinetic}\;\mathrm{equation}$$
(3)$$r_{i}=ke^{\frac{-E}{RT}}\alpha^{n_{1}}{\left(1-\alpha \right)}^{n_{2}}\;\mathrm{Proto}-\mathrm{kinetic}\;\mathrm{equation}$$
where *e* is the activation energy of moisture capacity, *k* is the pre-exponential factor, *r_i_* is the moisture capacity reaction rate, *R* is the ideal gas constant, *α* is the degree of conversion of moisture desorption reaction or stage, and *n*_i_
*is* the reaction orders of moisture capacity (i = 0, 1, 2) [19,22]. We developed a moisture capacity kinetic evaluation of *T. fuciformis* polysaccharides that includes the moisturizing characteristics, such as the reaction order, enthalpy of the moisture capacity (∆*H*), mass loss (mass%), and activation energy of moisture capacity, which could be applied in the design of processing conditions for cosmetics and biomaterials.

### 3.9. Statistical Analysis

All the measurements were determined in triplicate through all steps for each component. The experimental data were subjected to an analysis of variance (ANOVA) for a completely random design (CRD) using Statistical Analysis System (SAS Institute, Inc., Cary, NC, USA, 2009) to determine the least significant difference among means at the level of *p* < 0.05.

## 4. Conclusions

Overall, the study conducted *T. fuciformis* tests under various drying processes of freeze-drying, cold air-drying, and hot air-drying, observing that the freeze-dried sample had a high WI value for the cleanest white specimen and the best thermal decomposition properties for less residues than the others. We also determined that the freeze-dried sample had the lowest extraction rate and moisture content of polysaccharide, but had an excellent moisturizing ability in all samples. In addition, when comparing the moisture adsorption kinetic parameters of *T. fuciformis* polysaccharides of various drying processes by DSC and TGA thermal analyses and two types of Erofeev’s and proto-kinetic equations, the proto-kinetic equation could appropriately predict the moisture desorption capacity of the thermal effect under a closed system in air and nitrogen conditions. Moreover, we obtained the moisture adsorption capacity results of *T. fuciformis* polysaccharides by various drying processes; the desorption of moisture is more difficult in an air atmosphere, but is easier under limited closed conditions. The results and experience of this research will be applied to drying methods of natural material, evaluating fresh food processing, and preservation technology, and cosmetic additives and biomedical materials will also be included in future work.

## Figures and Tables

**Figure 1 molecules-24-02856-f001:**
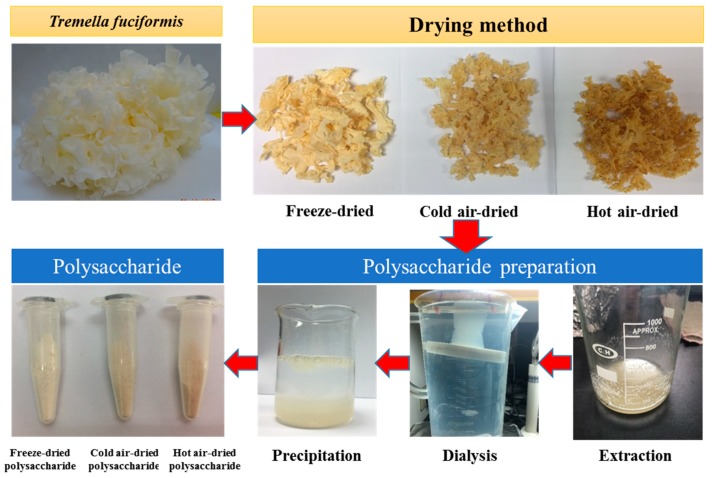
The flow diagram of the *T. fuciformis* sample and polysaccharide preparation.

**Figure 2 molecules-24-02856-f002:**
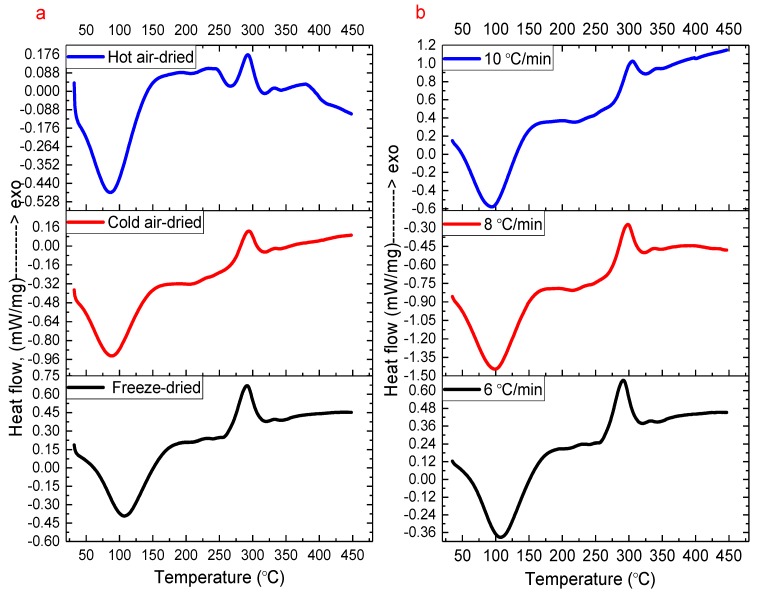
Differential scanning calorimetry (DSC) thermal curves for (**a**) the *T. fuciformis* polysaccharides dried by various processes at the selected scanning rate of 6 °C/min and (**b**) freeze-drying with the multiple scanning rates of 6, 8, and 10 °C/min.

**Figure 3 molecules-24-02856-f003:**
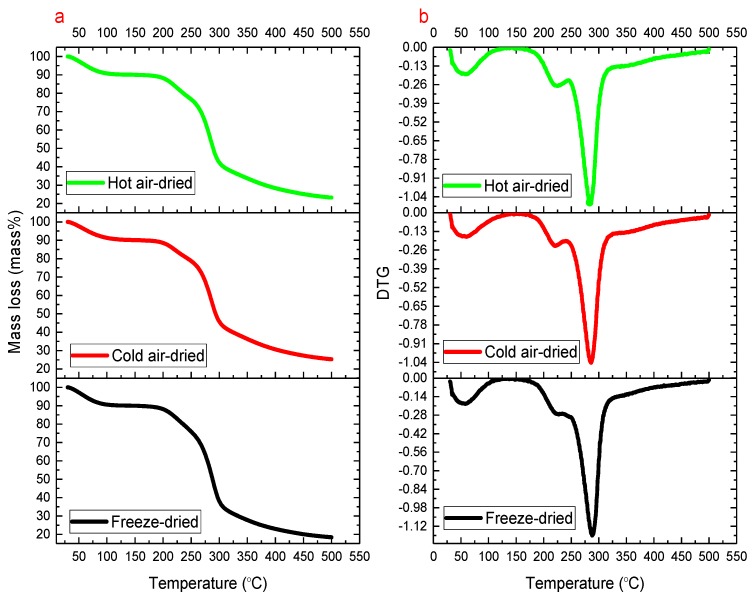
(**a**) Thermogravimetry (TGA) and (**b**) differential thermogravimetric (DTG) curves of *T. fuciformis* polysaccharides dried by various processes under nitrogen at the selected heating rate of 6 °C/min (this figure only presents partial results, and the complete analysis results were obtained with multiple heating rates of 6, 8, and 10 °C/min).

**Figure 4 molecules-24-02856-f004:**
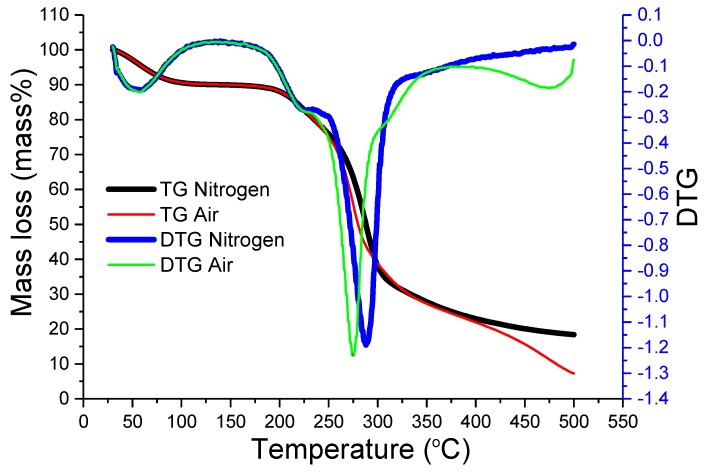
Thermogravimetry (TGA) and differential thermogravimetric (DTG) curves of the *T. fuciformis* polysaccharides obtained from freeze-drying in nitrogen and air atmospheres, respectively, at the selected heating rate of 6 °C/min (the complete analysis results were obtained with multiple heating rates of 6, 8, and 10 °C/min).

**Figure 5 molecules-24-02856-f005:**
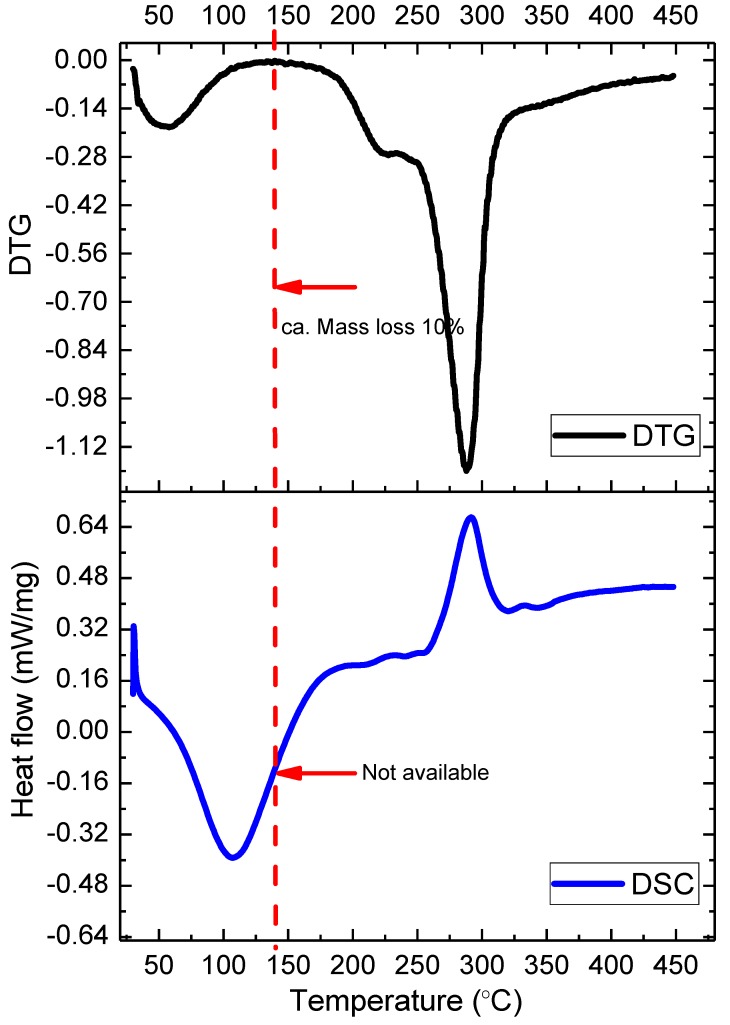
Comparisons of the differential scanning calorimetry (DSC) and differential thermogravimetric (DTG) curves of the *T. fuciformis* polysaccharides obtained from freeze-drying under a nitrogen condition at the selected heating rate of 6 °C/min.

**Figure 6 molecules-24-02856-f006:**
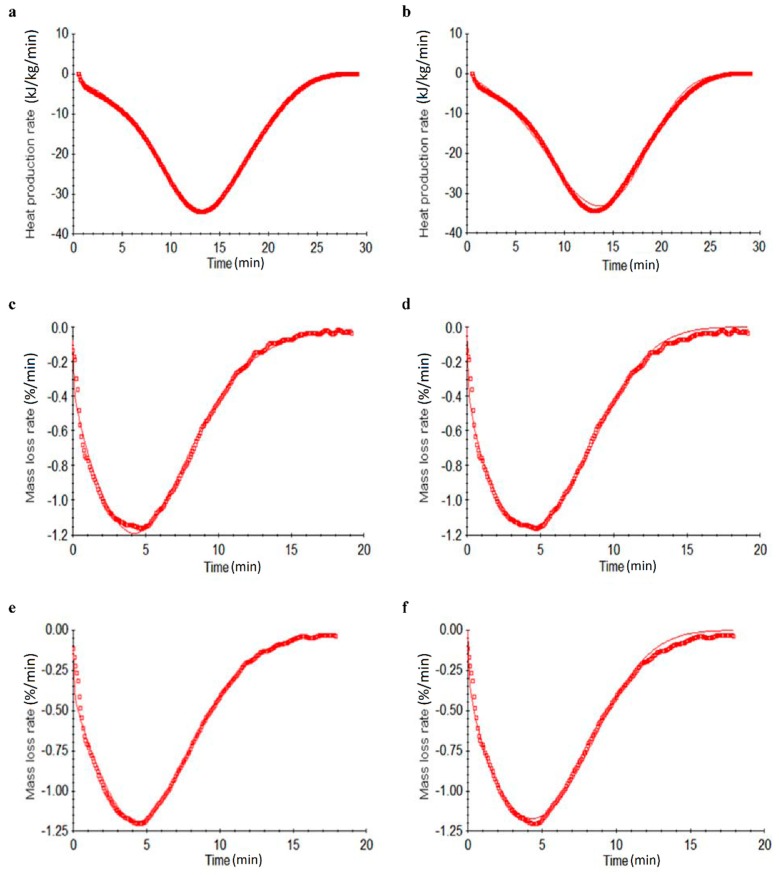
Adsorption kinetics of differential scanning calorimetry (DSC) (**a**,**b**) and differential thermogravimetric (DTG) (**c**,**d** under a nitrogen atmosphere; **e**,**f** under an air atmosphere) curves by comparing proto-kinetic (**a**,**c**,**e**) and Erofeev’s (**b**,**d**,**e**) equation simulation analyses at the selected heating rate of 6 °C/min (hollow block is experimental data and — line is simulation results) (here, only the heating rate of 6 °C/min analysis is presented, but the complete analysis results also involved the various heating/scanning rates of 8 and 10 °C/min, and then compared all the results).

**Table 1 molecules-24-02856-t001:** Color evaluation of *T. fuciformis* by various drying processes.

Sample	*L** Value	*a** Value	*b** Value	WI Value
Freeze dried	79.33 ± 0.09 ^a^	7.79 ± 0.06 ^c^	16.02 ± 0.04 ^c^	72.71 ± 0.12 ^a^
Cold air dried	72.39 ± 0.17 ^b^	9.48 ± 0.37 ^b^	19.86 ± 0.64 ^b^	64.70 ± 0.72 ^b^
Hot air dried	72.59 ± 1.05 ^b^	10.23 ± 0.27 ^a^	22.97 ± 0.19 ^a^	62.81 ± 0.99 ^c^

Legend: *L**, lightness; *a**, greenness/redness; *b**, blueness/yellowness; WI, whiteness index. All values are presented as means ± SD (*n* = 3). Means with different letters within a column differ significantly (*p* < 0.05).

**Table 2 molecules-24-02856-t002:** Yield, molecular weight, and moisture content of *T. fuciformis* polysaccharides obtained by various drying processes.

Sample	Yield (%)	Molecular Weight (Da)	Moisture Content (%)
Freeze dried	15.35	2.06 × 10^7^	11.61
Cold air dried	15.60	2.41 × 10^7^	13.05
Hot air dried	16.10	2.34 × 10^7^	12.65

**Table 3 molecules-24-02856-t003:** Results of differential scanning calorimetry (DSC) tests of various drying processes with scanning rates of 6, 8, and 10 °C/min in a temperature rise range chosen from 30 to 450 °C.

Sample	Mass (mg)	Heating rate (°C/min)	*enT*_o_ (°C)	*enT*_p_ (°C)	*en*Δ*H* (kJ/kg)	*exT*_o_ (°C)	*exT*_p_ (°C)	*ex*Δ*H*_d_ (kJ/kg)
Freeze dried	1.12	6	56.27	107.84	362.70	265.81	291.16	108.70
	1.13	8	42.98	99.30	304.20	274.10	297.54	72.73
	1.02	10	42.67	95.25	299.50	282.13	303.90	60.61
Cold dried	1.12	6	41.67	89.42	321.90	269.13	293.19	79.98
	1.20	8	41.59	91.27	331.30	277.11	299.23	52.31
	1.06	10	42.09	88.83	279.50	282.25	305.50	54.64
Hot dried	1.02	6	41.68	88.32	253.50	273.47	292.77	35.91
	1.60	8	43.59	94.36	350.60	274.00	296.25	79.51
	1.05	10	46.47	90.99	221.20	280.79	302.75	48.55

Legend: *enT*_o_, onset temperature of moisture adsorption; *enT*_p_, peak maximum temperature of moisture adsorption; *en*Δ*H*, enthalpy of the moisture capacity; *exT*_o_, onset temperature of exothermic reaction; *exT*_p_, peak maximum temperature of exothermic reaction; *ex*Δ*H*_d_, heat of thermal decomposition.

**Table 4 molecules-24-02856-t004:** Results of thermogravimetry (TGA) analyses for various drying processes from 30 to 500 °C with multiple heating rates of 6, 8, and 10 °C/min.

Sample	heating Rate (°C/min)	Nitrogen				Air			
		Mass (mg)	md*T*_p_ (°C)	td*T*_p_ (°C)	Mass Loss (mass%)	Mass (mg)	md*T*_p_ (°C)	td*T*_p_ (°C)	Mass Loss (mass%)
Freeze dried	6	6.02	57.50	288.11	−81.58	6.02	56.00	274.00	−92.70
	8	6.01	59.87	290.27	−80.84	6.01	60.93	279.60	−94.50
	10	6.00	63.50	296.00	−80.37	6.00	62.00	296.00	−93.28
Cold dried	6	6.00	60.00	286.00	−74.72	6.01	58.00	286.00	−75.54
	8	6.01	59.87	289.20	−76.74	6.01	64.13	281.73	−89.70
	10	6.00	65.50	294.00	−76.10	6.04	68.50	284.00	−88.07
Hot dried	6	6.01	57.50	284.00	−76.79	6.01	58.00	274.00	−91.83
	8	6.02	64.13	288.13	−77.60	6.02	59.87	279.60	−88.37
	10	6.00	64.50	292.00	−77.38	6.02	61.50	282.00	−87.95

Legend: md*T*_p_, peak temperatures of differential thermogravimetric curves of moisture desorption; td*T*_p_, peak temperatures of differential thermogravimetric curves of thermal decomposition.

**Table 5 molecules-24-02856-t005:** Moisture capacity kinetic simulation results of differential scanning calorimetry (DSC) (closed system) analyses for various drying processes.

Sample		Heating rates (°C/min)					
		6		8		10	
	Parameter	Erofeev	Proto	Erofeev	Proto	Erofeev	Proto
Freeze dried	ln(*k*_0_)/ln(1/s)	1.0 × 10^−6^	5.9911	1.0 × 10^−7^	5.9900	1.0 × 10^−9^	6.2444
	*E*_a_(kJ/mol)	18.2713	36.8027	16.7371	34.8600	15.6216	34.3314
	*n/n* _1_	0.3944	0.1663	0.4008	0.1950	0.4310	0.2038
	*n* _2_	N/A	1.2276	N/A	1.1647	N/A	1.1601
	Δ*H*(kJ/kg)	−412.7941	−418.0235	−333.7040	−336.6741	−336.2525	−338.0239
Cold dried	ln(*k*_0_)	1.0 × 10^−12^	8.8783	11.0162	0.0605	10.9169	2.8564
	*E* _a_	18.2685	46.0057	50.5996	16.5046	47.6868	23.6190
	*n/n* _1_	0.3951	0.0122	1.3943	0.4948	1.5242	0.3356
	*n* _2_	N/A	1.3537	N/A	0.8450	N/A	1.1172
	Δ*H*	−412.0653	−418.8865	−318.4601	−307.3842	−377.7577	−372.8248
Hot dried	ln(*k*_0_)	1.5744	5.3957	1.0 × 10^−8^	4.4862	1.0 × 10^−16^	3.9560
	*E* _a_	22.0218	33.1946	16.3484	29.6631	15.3977	26.4318
	*n/n* _1_	0.2160	0.0758	0.4104	0.2650	0.4332	0.4225
	*n* _2_	N/A	1.1473	N/A	1.0986	N/A	1.1435
	Δ*H*	−377.8556	−381.3110	−398.3055	−401.7383	−280.5788	−282.0463

Legend: ln(*k*_0_), natural logarithm for the pre-exponential factor of reaction; *E*_a_, activation energy of moisture capacity; *n*_i_, reaction order of the reaction or ith stage (dimensionless) i = 0, 1, 2; Δ*H*, Enthalpy of the endothermic reaction by simulation, moisture mass loss by kinetic simulation; Erofeev, Erofeev’s kinetic equation; Proto, proto-kinetic equation; N/A, not applicable.

**Table 6 molecules-24-02856-t006:** Adsorption kinetic simulation results of thermogravimetry (TGA) (under nitrogen and air atmospheres) analyses for various drying processes.

Sample Atmospheres Parameter			Heating rates (°C/min)					
			6		8		10	
			Erofeev	Proto	Erofeev	Proto	Erofeev	Proto
Freeze dried	Nitrogen	ln(*k*_0_) (ln(1/s))	1.0 × 10^−8^	6.8139	1.0 × 10^−8^	7.6747	1.0 × 10^−10^	9.7388
		*E*_a_ (kJ/mol)	15.2080	33.1840	11.8237	32.7078	14.1710	40.8807
		*n/n* _1_	0.2437	0.1900	0.3164	0.1758	0.3261	0.1563
		*n* _2_	N/A	1.4639	N/A	1.3277	N/A	1.4953
		DM (mass%)	−9.8306	−10.0380	−10.0056	−10.1331	−9.8335	−9.9987
Cold dried		ln(*k*_0_)	18.3377	4.7873	3.0 × 10^−8^	8.9333	3.2 × 10^−8^	1.9764
		*E* _a_	64.7625	28.1377	15.7559	39.3054	14.9599	19.7081
		*n/n* _1_	2.7817	0.2221	0.1584	0.1354	0.2556	0.3243
		*n* _2_	N/A	1.6567	N/A	2.1329	N/A	1.4310
		DM	−10.3558	−10.1801	−10.4496	−11.1291	−9.4918	−10.1712
Hot dried		ln(*k*_0_)	1.0 × 10^−6^	7.9012	1.0 × 10^−10^	8.7524	1.0 × 10^−8^	15.5255
		*E* _a_	15.3605	36.3266	12.0932	36.0103	14.4254	57.3172
		*n/n* _1_	0.2440	0.1720	0.3100	0.1444	0.3158	2.6 × 10^−3^
		*n* _2_	N/A	1.6172	N/A	1.4510	N/A	1.9482
		DM	−9.7767	−10.0970	−9.7811	−9.9547	−9.6731	−9.9811
Freeze dried	Air	ln(*k*_0_)	1.0 × 10^−6^	9.2593	6.3 × 10^−8^	8.2717	0.1711	7.7847
		*E* _a_	15.0903	39.9022	14.7013	37.2788	14.8925	35.8396
		*n/n* _1_	0.2657	0.1387	0.2818	0.1202	0.2851	0.1321
		*n* _2_	N/A	1.5232	N/A	1.3931	N/A	1.3813
		DM	−9.7068	−9.9269	−9.8599	−10.0177	−9.8607	−10.0220
Cold dried		ln(*k*_0_)	1.0 × 10^−5^	7.1195	1.0 × 10^−7^	12.0832	2.4 × 10^−10^	6.1735
		*E* _a_	16.0985	34.5655	16.7190	49.2521	15.1980	31.6670
		*n/n* _1_	0.2016	0.2193	0.1780	0.0317	0.2109	0.1742
		*n* _2_	N/A	1.9456	N/A	2.1931	N/A	1.7107
		DM	−9.7935	−10.4593	−10.3370	−10.9423	−10.1435	−10.6442
Hot dried		ln(*k*_0_)	1.0 × 10^−3^	11.2793	1.0 × 10^−7^	10.3135	0.3072	6.7677
		*E* _a_	15.3679	45.5960	14.8336	42.8635	15.3190	32.9951
		*n/n* _1_	0.2304	0.0735	0.2535	0.0659	0.2716	0.1555
		*n* _2_	N/A	1.7008	N/A	1.5940	N/A	1.3461
		DM	−10.6665	−10.9321	−10.2557	−10.4841	−10.2999	−10.4632

Legend: ln(*k*_0_), natural logarithm for the pre-exponential factor of reaction; *E*_a_, activation energy of moisture capacity; *n*_i_, reaction order of the reaction or *i*th stage (dimensionless) i = 0, 1, 2; DM, moisture mass loss by kinetic simulation; Erofeev, Erofeev’s kinetic equation; Proto, proto-kinetic equation; N/A, not applicable.

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
