# Peer review of "Differences in the Moisture Capacity and Thermal Stability of Tremella fuciformis Polysaccharides Obtained by Various Drying Processes"

_molecules, 2019, doi:10.3390/molecules24152856_

Round 1
Reviewer 1 Report
Manuscript entitled ‘Differences of the moisture capacity and thermal stability of Tremella fuciformis polysaccharides by various drying processes’ aimed at the evaluation of 3 drying methods on the selected physical properties of polysaccharides from mushroom. Unfortunately, there are points that should be consider before the possibility to accept the manuscript.
1) The concept of drying is not proper. Authors selected three drying methods: freeze-drying pointing wrongly the temperature (the process might be performed only at -40°C, the temperature of heating plate was not mentioned – whole the process cannot be performed only at – 40°C), cold air drying and hot air drying. They select the drying time for each sample. The duration of the processes was influencing the moisture content, why such long processes were used? This is obvious that when the freeze drying last for 96h, the obtained product had the lowest moisture content.
2) Regarding the description of the drying processes: i.e. line 111 ‘dried by freezing’ is wrong. Dried by sublimation – this is the process name. By freezing the evaporation of water is extremely slow when compared to freeze-drying process.
3) Authors repeat the results from tables in the text. There is no comparison when just the numbers are indicated. For readers is difficult to see the proper discussion.
4) The is no statistical analysis performed. Authors did not even mention the number of repetitions when performing the drying, or determination of selected parameters.
5) The conclusions are not precise. For example, the last sentence is not a conclusion, it is a description what Authors did – not acceptable in the scientific
6) what does it mean (line 29): contains rich carbohydrates???
Reviewer 2 Report
Paper titel: Differences of Moisture Capacity and Thermal Stability of Tremella fuciformis Polysaccharides by Various Drying Processes described by Lin et al.
In this study, Lin et al., characterized the influence of different drying techniques like hot air, low temperature and freeze-drying on moisture capacity and termal stability of Tremella fuciformis polysaccharides. They are also tested DSC, polysaccharide molecular distribution analysis, colour, The entire manuscript is well-designed and well-written. Data are organized in a scientific order.
Additionally comments:
The aim of this study show present much more clear.
How many drying repetition was done for each sample ?
In conclusion section Authors should add some information about perspective to use this polysaccharide, i.e. which drying techniques should be the best for analysis.
Some part of the conclusion are the same and present information which should present in Abstract.
Additionally shoud be deleted: “we …..” i.e. “we obtain” etc. in whole manuscript
Round 2
Reviewer 1 Report
Dear Authors,
you have stated that the "We have rewritten the description in the current text" when it concerns the statistical analysis. Actually, there is no change as i.e. Table 1 is still missing the repetitions (i.e. standard deviation) and the statistical analysis.
The answer provided: P13, Line 239-241: The cold air drying and at 18 °C for 96 h. The hot air drying and freeze drying at 50 °C and –40°C for 72 h, respectively. Cold air drying takes the longest time because water is not easily removed at normal pressure and low temperature.
is not an answer for the question. Reviewer know that drying at lower temperatures requires more time. The question is why the drying methods are so long? What was the final moisture content for each sample obtained by selected methods? The statement that below 11% wet basis is too general (one can be: 2% and the second 11%).
What does it mean: Fresh fruiting bodies? I believe that this will not be clear for the readers
In my opinion the conclusions were not rewritten as stated in the letter, just sentence has been removed. In my opinion, the conclusions should be rewritten.
